# An Injectable Chitosan-Based Self-Healable Hydrogel System as an Antibacterial Wound Dressing

**DOI:** 10.3390/ma14205956

**Published:** 2021-10-11

**Authors:** Xiaoyu Wang, Rijian Song, Melissa Johnson, Sigen A, Zhonglei He, Cameron Milne, Xianqing Wang, Irene Lara-Sáez, Qian Xu, Wenxin Wang

**Affiliations:** The Charles Institute of Dermatology, School of Medicine, University College Dublin, Dublin 4, Eircode D04 V1W8, Ireland; xiaoyu.wang2@ucdconnect.ie (X.W.); rijian.song@ucdconnect.ie (R.S.); melissa.johnson@ucdconnect.ie (M.J.); sigen.a@ucd.ie (S.A.); zhonglei.he@ucd.ie (Z.H.); cameron.milne@ucdconnect.ie (C.M.); xianqing.wang@ucdconnect.ie (X.W.); irene.lara-saez@ucd.ie (I.L.-S.)

**Keywords:** chitosan, hydrogel, self-healable, antibacterial, wound dressing

## Abstract

Due to their biodegradability and biocompatibility, chitosan-based hydrogels have great potential in regenerative medicine, with applications such as bacteriostasis, hemostasis, and wound healing. However, toxicity and high cost are problems that must be solved for chitosan-based hydrogel crosslinking agents such as formaldehyde, glutaraldehyde, and genipin. Therefore, we developed a biocompatible yet cost-effective chitosan-based hydrogel system as a candidate biomaterial to prevent infection during wound healing. The hydrogel was fabricated by crosslinking chitosan with dialdehyde chitosan (CTS–CHO) via dynamic Schiff-base reactions, resulting in a self-healable and injectable system. The rheological properties, degradation profile, and self-healable properties of the chitosan-based hydrogel were evaluated. The excellent antibacterial activity of the hydrogel was validated by a spread plate experiment. The use of Live/Dead assay on HEK 293 cells showed that the hydrogel exhibited excellent biocompatibility. The results demonstrate that the newly designed chitosan-based hydrogel is an excellent antibacterial wound dressing candidate with good biocompatibility.

## 1. Introduction

Wound healing is a physiological process that includes homeostasis, inflammation, proliferation, and tissue remodeling stages [1,2]. Wound infection can trigger the body’s immune response, lead to inflammation and tissue damage, and slow down the body’s healing process [3]. If not treated, wound infection may cause complications and require medical intervention [4]. The most serious local complication of infected wounds is related to wound healing, which can result in wound healing failure. Systemic complications include cellulitis, osteomyelitis, and septicemia. Therefore, wound infection is costly in terms of delayed healing and adverse effects on patient quality of life [5].

Antibiotics have been utilized widely to treat infections caused by microorganisms since their discovery in 1928. While antibiotics are beneficial to us, their overuse is contributing to the global crisis of multidrug-resistant microorganisms. Antibiotic resistance already accounts for around 13 million deaths per year [6]. The application of antibiotics to infectious wounds can lead to drug resistance, which is not conducive to wound healing. Consequently, wound dressings with antibacterial functions are increasingly considered excellent biomaterials for wound healing, due to their ability to reduce the risk of infection in partial- and full-thickness wound sites, as well as their biocompatibility. To better manage the wound, these wound dressings must be biocompatible and capable of effectively covering an irregular wound surface. Hydrogels with antibacterial activity, injectability, biocompatibility, and appropriate mechanical properties appear to meet most requirements [4]. Hydrogels are formed by crosslinking hydrophilic polymers, and the 3D networks combine the characteristics of wet wound healing with high water content. To date, a wide range of hydrogels with antibacterial functionality have been explored as biomaterials for wound healing [4,6,7]. According to current research on the development of antibacterial hydrogels, hydrogels composed of naturally derived antibacterial polymers are attracting significant attention. Chitosan (CTS) is a promising material with antibacterial activity and good biocompatibility.

CTS is a natural linear polysaccharide (Figure 1A) derived from chitin in the exoskeletons of crustaceans such as crab, shrimp, and crawfish by deacetylation [8]. CTS can accelerate wound healing due to its excellent biological activities, including inducing blood coagulation, fibroblast proliferation, and collagen deposition [4,9,10]. CTS in particular has excellent antibacterial activity, and is generally considered non-toxic toward mammalian cells [11]. Chemical crosslinking is a common method used in the preparation of chitosan-based biomaterials to improve their mechanical and physicochemical properties [12,13]. However, chitosan-based hydrogel crosslinking agents such as formaldehyde, glutaraldehyde, and genipin are associated with toxicity and high cost; these issues need to be solved [12,14,15,16,17,18]. Dialdehyde chitosan (CTS–CHO) is a crosslinker derived from chitosan that is characterized by low toxicity, abundant resources, and facile preparation [19,20,21]. CTS–CHO has been studied for its potential utility as a crosslinker and has demonstrated excellent performance [22,23].

Although chitosan-based hydrogels have been extensively studied, there have been no reports of hydrogels prepared by the crosslinking of CTS and CTS–CHO for antibacterial applications. In this study, an injectable, chitosan-based, self-healable hydrogel system was fabricated with CTS and dialdehyde chitosan (CTS–CHO) via dynamic Schiff-base reaction for antibacterial applications. CTS–CHO was prepared from CTS and sodium periodate via an oxidation reaction. The rheological properties, degradation profile, injectability, and self-healable properties of the CTS-based hydrogel system were evaluated. The results of the spread plate experiment showed that the CTS-based hydrogel possessed excellent antibacterial activity. A cytocompatibility test showed that the material and hydrogel were biosafe. 

## 2. Materials and Methods

### 2.1. Materials

Chitosan (from crustacean shells) was purchased from Sigma-Aldrich (Dublin, Ireland) (viscosity: 800–2000 cp; average molecular weight: 310,000–375,000 Da). Glacial acetic acid (97.5%), sodium periodate, ethylene glycol, lysozyme, LB broth, and LB agar were purchased from Fisher (Dublin, Ireland). Dialysis tubes (cut-off Mw: 8 kDa) were purchased from Spectrum Lab(Dublin, Ireland). The complete medium was purchased from Invitrogen. *E. coli* (Dublin, Ireland) and HEK 293 cell lines were purchased from American Type Culture Collection. Live/Dead Cell Viability Assay Kit and alamarBlue Cell Viability Reagent were purchased from BioSciences (Dublin, Ireland) and Sigma-Aldrich (Dublin, Ireland), respectively. All the chemicals were used directly unless noted. All aqueous solutions were prepared with deionized water.

### 2.2. Preparation and Characterization of Dialdehyde Chitosan

#### 2.2.1. Preparation of Dialdehyde Chitosan

An amount of 0.97 g of chitosan was dissolved in 100 mL of 0.1 mol/L acetic acid solution and was continuously stirred at room temperature for 24 h. An amount of 0.64 g of sodium periodate was added to the solution and stirred continuously in the dark for 24 h. Ethylene glycol (0.5 mL) was then added to the reaction solution and stirred constantly for 1 h to stop the reaction. The solution was dialyzed with dialysis tubes (molecular weight cut-off 8 kDa) against 5 L of deionized water 5 times, and the deionized water was changed every 4 h. The purified solution was filtered with filter paper. The dialysate was lyophilized for 3 days, and the CTS–CHO was obtained.

#### 2.2.2. Fourier Transform Infrared Spectrometry

The infrared spectra of chitosan and CTS–CHO were recorded on a Bluk FT-IR spectrometer (ALPHA) with an ATR accessory at room temperature. Samples were placed in contact with the ATR accessory in the sampling area. The resolution was set as 4 cm^−1^. A total of 32 scans were conducted for each sample. Spectra were recorded from 4000 cm^−1^ to 400 cm^−1^. All sample spectra were subjected to background subtraction. The sample area was wiped with alcohol and dried completely before the next test.

#### 2.2.3. Element Analysis

The technique used for element analysis was quantitative flash combustion. The analysis was conducted on an Exeter elemental analyzer (CE440, Coventry, United Kingdom). Before elemental analysis, CTS–CHO was freeze-dried for 3 days and kept in a desiccator. Elemental analysis should directly reflect the oxidation degree (*F_ox_*) according to the equation below:(1)$$\frac{N}{C}=\frac{F_{A}+\left(1-F_{A}\right)\cdot \left(1-F_{ox}\right)}{2F_{ox}+6}$$
where *F_A_* is the content of acetylated units, *N* is the percentage of nitrogen, and *O* is the percentage of oxygen.

#### 2.2.4. Bacteria Culture

*E. coli* was stored in 50% glycerin at −80 °C. A loopful of inoculum was streaked on the plates and cultured at 37 °C overnight. A colony was scraped and cultivated overnight in 10 mL of LB broth at 37 °C with a 120 rpm rotation. A 20 μL aliquot of bacteria suspension was added into 10 mL of LB broth, then incubated for 5 h at 37 °C with a 120 rpm rotation. The optical density of the bacterial suspension was then measured at 600 nm. Then, the bacterial suspension was diluted to 0.5 MacFarland concentration (bacteria concentration: 1.5 × 10^8^ CFU/mL) with LB broth.

#### 2.2.5. Agar Diffusion Assay

To determine the antibacterial activity of CTS–CHO, 10, 30, and 50 mg/mL CTS–CHO solutions were prepared by dissolving CTS–CHO in deionized water. Under sterile conditions, 200 μL bacterial suspension (1 × 10^6^ CFU/mL) of *E. coli* was spread onto the LB agar plates evenly. Then, 20 μL of the sample was dropped on a filter paper slice (diameter: 8 mm). The negative control was sterile deionized water instead of CTS–CHO. The plates were incubated at 37 °C for 24 h. The circular region where bacteria could not grow around the samples was considered the inhibition zone.

### 2.3. Preparation and Characterization of Chitosan-Based Hydrogel

#### 2.3.1. Preparation of Chitosan-Based Hydrogel

An amount of 0.5 g of CTS was dissolved in 20 mL of acetic acid aqueous solution (0.1 mol/L) and was continuously stirred at room temperature for 24 h. An amount of 0.5 g of CTS–CHO was dissolved in 10 mL of deionized water and vortexed for 30 min. CTS–CHO solution (8 mL) was added to 20 mL of CTS solution, and then the mixture was vortexed for 30 s to form the uniform hydrogel.

#### 2.3.2. Rheological Test

The viscoelastic properties of hydrogels were determined by a Discovery HR-2 Rheometer (TA Instrument, HR-2, New Castle, Delaware, USA) with a parallel-plate geometry (diameter: 8 mm) at 25 °C. The gap was 2 mm for all measurements. An amount of 200 μL of the hydrogel was placed on the plate, and deionized water was dropped around the sample to prevent water evaporation. First, a time sweep study was conducted at a frequency of 1.0 Hz and strain of 1% to determine the change of the storage modulus and loss modulus with time. A strain sweep was conducted at a constant frequency of 1.0 Hz in the strain range of 1% to 1000% to evaluate the linear viscoelastic properties of the hydrogels. Then, the hydrogels were treated with 500% and 1% strains. The changes of the storage modulus and loss modulus were recorded.

#### 2.3.3. Degradation Test

Amounts of 0.2 g of the hydrogels (*n* = 3) were immersed in 20 mL of PBS solution (pH 7.4) and lysozyme solution (2 mg/mL). Lysozyme solution was prepared by dissolving lysozyme in PBS solution (pH 7.4). The degradation test was carried out at 37 °C. At the pre-determined time points, the residual samples were dried with filter paper and weighed. The PBS solution or lysozyme solution was replaced with the newly prepared solution after weighing at each time point.

The degradation rate (*DR*) was defined by the following equation:(2)$$DR\%=\frac{\left(W_{0}-W_{t}\right)}{W_{0}}\times 100\%$$
where *W*_0_ is the initial weight of the hydrogel and *W_t_* is the weight of the residual hydrogel at pre-determined time points.

#### 2.3.4. Self-Healable and Injectable Properties

The stained hydrogel was placed in a syringe at room temperature for 1 h. Then, the hydrogel was injected in three individual pieces. The three stained pieces were placed together gently with full contact. The photos were taken after 1 h to estimate the self-healable property.

The hydrogel was prepared and placed in a syringe for 1 h. Then, the hydrogel was injected through a needle (22 G) to evaluate its injectable properties.

#### 2.3.5. Anti-Bacterial Property

CTS and CTS–CHO hydrogels (0.3 g per well) were placed in a 48-well plate. The hydrogel and the blank well (without hydrogel) were sterilized with 1 mL of 75% ethanol for 5 min, then washed with 1 mL of sterilized water 3 times, for 30 min each time. An amount of 100 µL of bacteria solution (1 × 10^5^ CFU/mL) was added to the hydrogel and the blank well surfaces, respectively. After incubation at 37 °C for 12 h, 900 µL of LB broth was added to the wells and incubated for 2 h at 37 °C to suspend the bacteria. Bacteria solution (100 µL) was coated on the surface of the LB agar plate. The plates were incubated upside down for 24 h at 37 °C.

### 2.4. In Vitro Cytotoxicity Assessment

#### 2.4.1. Cytotoxicity of Materials

The cytotoxicity of CTS and CTS–CHO was tested using HEK 293 cells by alamarBlue assay. The stock solution was prepared by dispersing CTS and CTS–CHO in a 25 mM sodium acetate buffer (pH 5.2) and was extracted at 37 °C with 5% CO_2_ for 24 h, respectively. Afterwards, the stock solutions were filtered for sterilization using 0.22 µm filters and diluted with complete medium (DMEM with 10% FBS and 1% penicillin) to the concentrations of 10, 5, 2.5, 1.25, 0.625 mg/mL, respectively. The HEK 293 cells were inoculated with a density of 1 × 10^4^ cells per well in a 96-well plate and cultured in complete medium at 37 °C with 5% CO_2_. After overnight incubation, the culture medium was changed to a series of CTS or CTS–CHO solutions (from 0.01 to 10 mg/mL, *n* = 4). Wells containing complete medium were used as controls. Cell viability was tested 24 h later. An amount of 20 μL of alamarBlue reagent was added to each well. After 1 to 4 h of incubation, the absorbance was measured at 570 nm with a plate reader.

#### 2.4.2. Cytotoxicity of Hydrogel

An amount of 1000 mg of the CTS/CTS–CHO hydrogel was immersed in 5 mL of complete medium and cultured in 5% CO_2_ at 37 °C for 24 h. Then, the supernatant was filtered for sterilization using a 0.22 μm filter to obtain the extraction stock medium solution. HEK 293 cells were inoculated with a density of 1 × 10^4^ cells/well in a 96-well plate and were cultured in complete medium in 5% CO_2_ at 37 °C. Then, the cell culture medium was changed to hydrogel extract solution (*n* = 4) after overnight incubation. Control wells containing complete medium were used as controls. Live/Dead cell staining assay was performed after 24 h and 48 h of incubation to represent the cytocompatibility of the CTS/CTS–CHO hydrogel. The Live/Dead stain reagent was pre-thawed before use, then 2 μL of calcein-AM (Component A) and 8 μL of ethidium homodimer-1 (Component B) were added into 4 mL of DPBS. The staining solution was obtained after mixing. At determined time points, the culture medium was removed and replaced with 100 µL of staining solution per well. PBS was used to gently wash away the stain from the well plates after incubation at 25 °C for 30 min. Images were taken under a fluorescence microscope.

## 3. Results and Discussion

The main objective of this study was to fabricate an injectable and self-healable chitosan-based hydrogel with antibacterial properties for use as a wound dressing. Herein, we developed a biocompatible and biodegradable hydrogel by covalently cross-linking chitosan with dialdehyde chitosan by imine linkages (Figure 1).

### 3.1. Preparation and Characterization of Dialdehyde Chitosan (CTS–CHO)

The structures of CTS and CTS–CHO were detected by FT-IR to confirm that the aldehyde groups were introduced into the structure of chitosan. The results are shown in Figure 1. The band at 3355 cm^−1^ is attributable to O-H, whereas the bands at 2870 cm^−1^ and 1150 cm^−1^ correspond to the stretching of C-H and C-O-C. The absorption bands at 1645 cm^−1^ and 1557 cm^−1^ are assigned to C-O of amide I and N-H of amide II, respectively. The bands at 1060 and 1025 cm^−1^ are ascribed to the stretching of C-O [12,24,25]. Comparisons between the FT-IR spectra of CTS–CHO and CTS revealed the presence of a characteristic absorption peak of aldehyde groups at 1723 cm^−1^ caused by sodium periodate oxidation. The characteristic absorption peak confirmed that CTS–CHO was prepared successfully [12,20,22].

The proportion of dialdehyde groups per 100 GlcN units is defined as the oxidation degree of CTS–CHO 20. The oxidation degree of CTS–CHO was calculated based on element analysis, and the result was 14.75%, as shown in Appendix A. This result is consistent with the published data [12,20].

As described in the literature, the minimal inhibitory concentration (MIC) of CTS against *E. coli* is 48 μg/mL [26]. The antibacterial activity of CTS–CHO against *E. coli* was assessed by an agar diffusion assay. The images of the bacterial growth inhibition profile and the diameter of the bacterial growth inhibition zone are shown in Appendix A, and in Appendix A. The inhibitory zones of CTS–CHO at the concentrations of 30 mg/mL and 50 mg/mL were 13.5 mm and 15.0 mm, respectively. The results suggest that CTS–CHO could significantly inhibit the growth of *E. coli,* indicating that chitosan retains its antibacterial activity after oxidation. A similar conclusion was reported in the literature [22]. The antibacterial activity of CTS–CHO is due to the positively charged amino group in its molecular structure. Therefore, CTS–CHO has the potential to be used as a biological crosslinking agent with antibacterial activity.

### 3.2. Preparation and Characterization of Chitosan-Based Hydrogel

As described previously, the chitosan-based hydrogel was developed by crosslinking CTS and CTS−CHO via Schiff-base reaction between amine and aldehyde groups, as shown in Figure 1B. The interactions between CTS and CTS−CHO in the reaction, such as hydrophobic sidechain aggregation and intermolecular and intra-molecular hydrogen bond formation, contribute to the self-healable and mechanical properties of the chitosan-based hydrogel [27].

CTS is insoluble in all solvents except aqueous acidic solutions [28,29]. To fabricate the hydrogel, we dissolved CTS (2.5% *w*/*v*) in 0.1 mol/L acetic acid aqueous solvent and stirred for 24 h at room temperature. The hydration of aldehyde groups in the molecular structure considerably increased CTS−CHO solubility. CTS−CHO can be dissolved in water within a certain concentration range. Therefore, we prepared 5% CTS−CHO solution by dissolving 0.5 g of CTS−CHO in 10 mL of deionized water and vortexing for 30 min. An amount of 8 mL of the CTS−CHO solution was then added to 20 mL of the CTS solution and vortexed to form the uniform hydrogel.

To evaluate the equilibrium moduli and gelation profile, a time sweep experiment was conducted. The gelation was evidenced by the gradual increase of G’, which achieved a maximum equilibrium value of 356 Pa within 30 min (Figure 2A). The strain sweep results showed that the linear-viscoelastic region of the hydrogel concerning strain was between 1% and 250% (Figure 2B). The storage modulus remained constant at strains ranging from 1% to 250%, indicating a mechanically robust viscoelastic property. Then, a step-strain amplitude experiment was carried out to determine the self-healing profile of the hydrogel. The 500% and 1% strains were alternately applied to the hydrogel for four cycles. As shown in Figure 2C, the viscoelastic property decreased significantly when the strain was increased to 500%. Additionally, the viscoelastic property recovered to its initial value when the strain was returned to 1%. The self-healing profile was consistent throughout the four cycles. The results indicate that the hydrogel possessed excellent thixotropic properties based on dynamic imine linkages [30,31].

The mechanisms of chitosan-based biomaterials degradation include both physical and chemical processes. Nonenzymatic hydrolysis and enzymatic degradation in vitro were conducted to assess the degradation profile of the hydrogels. The glycosidic bonds between polysaccharide units were hydrolyzed, but at a slower rate than enzymatic hydrolysis. The hydrogels retained 56% of their initial weights in 3 days in PBS solution at 7.4 pH (Figure 2D).

According to the literature, the concentration of lysozyme in different tissues ranges from 1 to 14 μg/mL. Lysozyme can degrade chitosan by cleaving the glycosidic bonds between polysaccharide units [32]. We conducted enzymatic degradation evaluation in 2 mg/mL lysozyme solution prepared with PBS at 7.4 pH. After 24 h, only 29.8% of that hydrogel’s weight remained. The hydrogel then degraded completely in 3 days (Figure 2D). Therefore, enzymatic degradation was the main mechanism of degradation for this chitosan-based hydrogel, while the nonenzymatic hydrolytic mechanism was a minor contributor.

Extrusion through a needle (22 G) indicated that the hydrogel possessed good injectability (Figure 3A). The image clearly shows that the hydrogel had thixotropic properties [33]. Self-healability is a beneficial characteristic of hydrogels for wound healing. After damage, the original condition of the hydrogel can be restored within a certain period without being affected by external factors. The self-healing properties of this chitosan-based hydrogel were observed visually. The stained hydrogels were placed into a syringe and separated into three pieces. The three stained pieces were placed together gently with full contact (Figure 3B). After contact for 1 h at room temperature, the hydrogel had self-healed due to the dynamic covalent imine linkages.

To determine the antibacterial activity of the chitosan-based hydrogels, a spread plate experiment was conducted. Images were taken after incubation for 24 h (Figure 4A). The number of colonies on the control plate was significantly higher than on the hydrogel plate. Due to the large number of positively charged amino groups in its molecular structure, chitosan has excellent antibacterial activity. CTS−CHO, the crosslinker obtained by the oxidation of chitosan, also contains many amino groups in its molecular structure.

The cytotoxicity of the CTS and CTS−CHO materials on HEK 293 cells was evaluated by the alamarBlue assay. Due to the insolubility of CTS under neutral conditions, we chose a sodium acetate buffer (25 mM, pH = 5.2) as the solvent to ensure that the experimental conditions for the two materials were the same. The stock solutions after extraction were diluted with complete media to the desired concentrations. The cell viability of CTS and CTS−CHO at various experimental concentrations was greater than 87%, indicating that CTS and CTS−CHO concentrations ranging from 0.625 to 10 mg/mL did not affect the viability of HEK 293 cells (Figure 4B). There were no statistical differences in the antibacterial activity between CTS and CTS–CHO. The results also demonstrated that the oxidation reaction of chitosan did not cause an increase in cytotoxicity.

The cytotoxicity of the chitosan-based hydrogel was investigated using a Live/Dead assay against HEK 293 cells by co-culturing the cells with the extracted solution of the hydrogel after 24 h and 48 h. The outcome was observed with a fluorescence microscope. Green fluorescence indicated living cells, whereas red fluorescence indicated dead cells. The majority of HEK 293 cells were viable after 24 h and 48 h following treatment with the hydrogel extract medium. There was no significant difference between the experimental group and the controls (Figure 4C,D), allowing us to conclude that the materials and the hydrogel were non-cytotoxic. This result also lays the foundation for the biomedical application of CTS–CHO and chitosan-based hydrogels.

## 4. Conclusions

In this study, we reported a biodegradable chitosan-based hydrogel system with excellent antibacterial activity, biocompatibility, injectability, and self-healable properties. The hydrogel was fabricated from chitosan and aldehyde chitosan. Aldehyde chitosan was demonstrated to be an excellent biomaterial crosslinker with antibacterial activity and water solubility. Moreover, the cell viability of aldehyde chitosan was greater than 87% in the concentration range from 0.625 to 10 mg/mL. The results of the Live/Dead assay indicated the negligible cytotoxicity of the formed hydrogel. Because of the antibacterial activity of chitosan and aldehyde chitosan in hydrogel networks, this hydrogel system significantly inhibited bacterial growth on contact. Meanwhile, the hydrogel possessed promising potential beneficial characteristics for wound healing, such as injectability and self-healability, which are awaiting further exploration. Overall, the novel combined hydrogel system reported in this study has strong potential to be an excellent candidate for antibacterial wound dressings.

## Data Availability

All the data are available within the manuscript.

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
