# Peer review of "An Injectable Chitosan-Based Self-Healable Hydrogel System as an Antibacterial Wound Dressing"

_materials, 2021, doi:10.3390/ma14205956_

Round 1

Reviewer 1 Report

The article by Wang et al. deals with the development of an injectable chitosan based hydrogel. The topic is suitable for the journal. However, there are already many articles on that topic published in the literature. Therefore, the novelty of the article should be clearly highlighted in the introduction. Other concerns should be addressed:

-More literature references should be provided in the introduction section, specially those dealing with the application of chitosan in wound dressing. See for instance:

Wound Healing Bionanocomposites Based on Castor Oil Polymeric Films Reinforced with Chitosan-Modified ZnO Nanoparticles. Biomacromolecules 2015, 16, 2631-2644.

-The discussion of the article it is very short. That part should be expanded via comparison with previous literature works in the field.

-Microscopic characterization of the developed microgel (via SEM, TEM, etc) should be provided.

-Water absorption and water vapour transsmition rate should be measured.

-Taking into account the targeted application, in vivo wound healing tests should be carried out.

Author Response

The article by Wang et al. deals with the development of an injectable chitosan based hydrogel. The topic is suitable for the journal. However, there are already many articles on that topic published in the literature. Therefore, the novelty of the article should be clearly highlighted in the introduction. Other concerns should be addressed.

Re: We really appreciate the reviewer’s comments and valuable suggestions, which helped us to improve our manuscript and research work. To address the raised concerns, we have listed our point-by-point response below.

In the introduction section, we revised the manuscript and supplemented some information on the novelty of our research. “Although chitosan-based hydrogels have been extensively studied, the hydrogel prepared by crosslinking CTS with CTS-CHO for antibacterial application has not been reported (line 62~64).”

1.More literature references should be provided in the introduction section, specially those dealing with the application of chitosan in wound dressing. See for instance:

Wound Healing Bionanocomposites Based on Castor Oil Polymeric Films Reinforced with Chitosan-Modified ZnO Nanoparticles. Biomacromolecules 2015, 16, 2631-2644.

Re: We appreciate the reviewer’s suggestion. The literature “Wound Healing Bionanocomposites Based on Castor Oil Polymeric Films Reinforced with Chitosan-Modified ZnO Nanoparticles” has been cited (reference 10).

2.The discussion of the article it is very short. That part should be expanded via comparison with previous literature works in the field.

Re: We appreciate the reviewer’s comment. The manuscript has been revised, and some discussion has been added to the discussion section (line 384~387).

3.Microscopic characterization of the developed microgel (via SEM, TEM, etc) should be provided.

Re: We appreciate the reviewer bringing this to our attention. The work reported in this manuscript is primarily a demonstration of the concept. Due to Covid-19 pandemic, access to electron microscopes is still limited in University College Dublin. As the reviewer suggested, we hope to conduct the microscopic characterization and further optimization in our follow up study.

4.Water absorption and water vapour transsmition rate should be measured.

Re: We thank the reviewer’s suggestion. The work reported in this manuscript is primarily a demonstration of the concept. For the same reason mentioned in comment 4, we currently have limited access to relevant instruments and hope to characterize this material further in a follow-up study.

5.Taking into account the targeted application, in vivo wound healing tests should be carried out.

Re: We thank the reviewer for pointing this out. The work reported in this manuscript is primarily a demonstration of the concept. In vivo tests are of particular interest to us, and we have planned animal experiments in a later study to confirm the in vivo wound healing effect. The study will be reported in the future.

Reviewer 2 Report

  1. all figures must be presented with a better resolution
  2. the conclusion needs to be revised, indicating concrete results of the study 

Author Response

We really appreciate the reviewer’s comments and valuable suggestions which helped us to improve our manuscript and research work. To address the raised concerns, we have listed our point-by-point response below.

1.all figures must be presented with a better resolution

Re: We thank the reviewer’s suggestion. The resolution of all figures have been adjusted.

2.The conclusion needs to be revised, indicating concrete results of the study

Re: We appreciate the reviewer’s comments. The conclusion has been revised (line 341~347).

Reviewer 3 Report

The paper is well written but I have some minor concerns that the author needs to address before publishing.

Line 165 166 „The three stained pieces were placed 165together and compressed slightly to ensure intimate contact.”  What strength?. The word intimate does not fit here. Incorrect scope of meaning of adjective.

Line 172 “blank well” Need to clarify what was in a blank well

Toxicity experiments are performed on the soluble fraction of the material. Before conducting the experiment, the sample is filtered and some of the material may settle on a filter. How much material was actually left on the filter.  Did not the authors try to seed cells on complete gels (previously incubated in medium).This would give a more realistic picture. The authors should comment on why the experiment was done this way.

HEK 293 is a very unfortunate line for research in regenerative medicine.The author should indicate why he used this line.

Author Response

The paper is well written but I have some minor concerns that the author needs to address before publishing.

Re: We really appreciate the reviewer’s comments and valuable suggestions which helped us to improve our manuscript and research work. In order to address the raised concerns, we have listed our point-by-point response below.

1.Line 165 166 „The three stained pieces were placed 165together and compressed slightly to ensure intimate contact.” What strength?. The word intimate does not fit here. Incorrect scope of meaning of adjective.

Re: We appreciate the reviewer’s comments. The three stained pieces were placed together gently with full contact. Then the hydrogel self-healed. We have revised the description as “The three stained pieces were placed together gently with full contact.”(line 158~159)

2.Line 172 “blank well” Need to clarify what was in a blank well.

Re: The term ‘blank well’ refers to the absence of hydrogel in the well. To eliminate distractions, we sterilized the empty well using the same method as the hydrogel, namely sterilized with 1 mL of 75% ethanol for 5 minutes, then washed with 1 mL of sterilized water 3 times, 30 minutes each time. The detailed description was supplemented in 2.3.5 section (line 165).

3.Toxicity experiments are performed on the soluble fraction of the material. Before conducting the experiment, the sample is filtered and some of the material may settle on a filter. How much material was actually left on the filter. Did not the authors try to seed cells on complete gels (previously incubated in medium). This would give a more realistic picture. The authors should comment on why the experiment was done this way.

Re: We thank the reviewer’s comments. Aldehyde chitosan is soluble in water, whereas chitosan is insoluble in water. To eliminate solvent interference, we chose 25 mM sodium acetate buffe to dissolve the materials. Aldehyde chitosan was completely soluble in 25 mM sodium acetate buffer. Chitosan is a natural polymer with excellent biocompatibility. In this study, the cytotoxicity evaluation of chitosan is mainly used to make a contrast with aldehyde chitosan. Therefore, the toxicity results of aldehyde chitosan should be reliable. We chose the method for cytotoxicity assessment of hydrogel based on ISO 10993-5 and hope it is sufficient and reliable to demonstrate the biocompatibility of the hydrogel. Seeding cells on the complete gels or the 3D culture of cells in the gel will be attractive for our follow-up study to explore the application of this material. We thank the reviewer again for the suggestion.

4.HEK 293 is a very unfortunate line for research in regenerative medicine. The author should indicate why he used this line.

Re: We thank the reviewer for this suggestion. To the our best knowledge, HEK 293 cell line is more sensitive to the extracellular environment than the epidermal cell line, such as fibroblast. Therefore, we chose HEK293 cells and found that the low toxicity of hydrogel presented in our study toward HEK 293 cells, making the hydrogel a promising biocompatible candidate. In our follow up study, we hope to test the wounding efficacy with further cell models, ex vivo models and in vivo models.

Reviewer 4 Report

The paper "An injectable Chitosan-based self-healable hydrogel system as 2 an antibacterial wound dressing" by Wang et al. was submitted to MDPI Materials Journal. 

The authors present well description of the topic in the introduction section. 

Further characterization is advisable before accepting the work for publications. 

  • SEM/cryo-SEM characterization of the obtained hydrogel 
  • Surface charge characterization can offer further details of the interaction of the hydrogel with other surfaces and cells.
  • Have the authors considered to use this material for coating of medical devices?
  • IR spectra would be nice to be seen in the main draft. 
  • Resolution of figure 1 should be further enhanced. 
  • TGA would provide further information of the chemical composition of the material.

I suggest major revisions before further consideration.  

Author Response

The paper "An injectable Chitosan-based self-healable hydrogel system as an antibacterial wound dressing" by Wang et al. was submitted to MDPI Materials Journal.

The authors present well description of the topic in the introduction section.

Further characterization is advisable before accepting the work for publications.

We really appreciate the reviewer’s comments and valuable suggestions which helped us to improve our manuscript and research work. To address the raised concerns, we have listed our point-by-point response below.

1.SEM/cryo-SEM characterization of the obtained hydrogel

Re: The work reported in this manuscript is primarily a demonstration of the concept. Due to Covid-19 pandemic, access to electron microscopes is still limited in University College Dublin. As the reviewer suggested, we hope to conduct microscopic characterisation and further optimisation in our follow-up study.

2.Surface charge characterization can offer further details of the interaction of the hydrogel with other surfaces and cells.

We thank the reviewer’s suggestion. The work reported in this manuscript is primarily a demonstration of the concept. For the same reason mentioned in comment 1, we currently have limited access to relevant instruments and hope to characterize this material further in a follow-up study.

3.Have the authors considered to use this material for coating of medical devices?

Re: We thank the reviewer for this suggestion. We have already studied the antibacterial application of CTS-CHO and made some progress. This material is suitable for coating applications due to its antibacterial activity and water solubility. The study will be reported in the future.

4.IR spectra would be nice to be seen in the main draft.

Re: We appreciate the reviewer’s comments. The IR spectra has been moved from the supporting information to the main manuscript (Figure 1).

5.Resolution of figure 1 should be further enhanced.

Re: The resolution of figure has been adjusted (Figure 2).

6.TGA would provide further information of the chemical composition of the material.

Re: We thank the reviewer’s suggestion. Due to Covid-19 pandemic, TGA is still not available in University College Dublin. We will conduct compositional analysis with TGA when the situation improves to characterize the material further. The study will be reported in the future.

Round 2

Reviewer 1 Report

The article has been significantly improved. Can be accepted for publication now.

Reviewer 3 Report

The authors have satisfactorily revised the paper and addressed all of the reviewer's comments. The paper in this form may be published.

Reviewer 4 Report

The authors have addressed significant changes on the manuscript, making it suitable for acceptance.